# Keystone pathobionts associated with colorectal cancer promote oncogenic reprograming

**Josh Jones, Qiaojuan Shi, Rahul R. Nath, Ilana L. Brito** *

Meinig School for Biomedical Engineering, Cornell University, Ithaca, NY, United States of America

* ibrito@cornell.edu

**Data Availability Statement:** The scRNA-seq data have been deposited in the GEO database (https://www.ncbi.nlm.nih.gov/geo/) under accession code GSE220136 (https://www.ncbi.nlm.nih.gov/geo/query/acc.cgi?acc=GSE220136).

## Abstract

*Fusobacterium nucleatum* (Fn) and enterotoxigenic *Bacteroides fragilis* (ETBF) are two pathobionts consistently enriched in the gut microbiomes of patients with colorectal cancer (CRC) compared to healthy counterparts and frequently observed for their direct association within tumors. Although several molecular mechanisms have been identified that directly link these organisms to features of CRC in specific cell types, their specific effects on the epithelium and local immune compartment are not well-understood. To fill this gap, we leveraged single-cell RNA sequencing (scRNA-seq) on wildtype mice and mouse model of CRC. We find that Fn and ETBF exacerbate cancer-like transcriptional phenotypes in transit-amplifying and mature enterocytes in a mouse model of CRC. We also observed increased T cells in the pathobiont-exposed mice, but these pathobiont-specific differences observed in wildtype mice were abrogated in the mouse model of CRC. Although there are similarities in the responses provoked by each organism, we find pathobiont-specific effects in Myc-signaling and fatty acid metabolism. These findings support a role for Fn and ETBF in potentiating tumorigenesis via the induction of a cancer stem cell-like transit-amplifying and enterocyte population and the disruption of CTL cytotoxic function.

## Introduction

Colorectal cancer (CRC) is caused by both genetic mutations and aberrant features of the gut microbiome. Specifically, two organisms, *Fusobacterium nucleatum* (Fn) and enterotoxigenic *Bacteroides fragillis* (ETBF), are commonly enriched in the gut microbiomes of CRC patients [1–7] and exacerbate intestinal tumor formation in CRC mouse models [5,8]. Although a handful of molecular mechanisms have been identified that directly link these organisms with oncogenic pathways, less is known about how they affect distinct cell types within the intestinal compartment.

Fn was originally identified as an oral pathobiont due to its role in subgingival and periodontal disease [9,10], more recent studies find that Fn is associated with a number of cancers, including esophageal cell carcinoma [11,12], breast cancer [13], and most extensively with CRC [2,7,8,14–17]. Within CRC patients, Fn is spatially enriched in both adenomas and

**Funding:** This work was supported in part by a grant from the National Cancer Institute (1R33CA235302-01A1). J.J. was funded by a fellowship from the Center for Vertebrate Genomics at Cornell University. I.L.B. was funded by the National Heart, Blood and Lung Institute (1DP2HL141007), a Research Fellowship from the Pew Charitable Trusts and a Fellowship from the David and Lucille Packard Foundation. The funders had no role in study design, data collection and analysis, decision to publish, or preparation of the manuscript.

**Competing interests:** The authors have declared that no competing interests exist.

adenocarcinomas [7,14,16–18]. Fn is often present on CRC tumor tissue and this is linked to its expression of several adhesins, including FadA [19,20], and Fap2, the latter of which binds to the sugar residue, Gal-GalNAc [21,22], overexpressed on CRC tumors [23]. In addition to these associations, Fn has been shown to play a causative role in neoplastic transformation, with several recognized mechanisms. *Fusobacterium*-specific effector protein Fap2 interacts with TIGIT (T cell immunoreceptor with immunoglobulin and ITIM domain), a potent mediator of immunosuppression, leading to reduced natural killer cell and cytotoxic T cell mediated cytotoxicity [24]. Additionally, in *in vitro* and *in vivo* models of CRC, including the commonly used $Apc^{Min/+}$ mouse model, Fn protein FadA has been shown to bind to host cells and promote host DNA damage [25]. This consequently induces beta-catenin and Wnt signaling [26] and annexin A1 expression [27], which together trigger intestinal cell proliferation [8,28].

Under homeostatic conditions, non-toxigenic *B. fragilis* strains are highly prevalent gut commensals. However, certain *B. fragilis* strains express *B. fragilis* toxin (Bft) and are a common clinicopathological feature in inflammatory bowel disease (IBD) [29–31], diarrheal disease [32], and CRC [3–6]. ETBF has been shown to play a causal role in murine models of CRC. Specifically, Bft acts as a zinc-dependent metalloprotease that degrades E-cadherin, leading to aberrant signaling by beta-catenin and c-myc, both of which support enterocyte growth and proliferation [5,33–36]. Furthermore, ETBF exposure elicits robust pro-tumorigenic IL-17 production and Th17 and T regulatory cell responses [37–40], further establishing a pro-oncogenic role for this pathobiont.

To investigate the effects of Fn and ETBF on host intestinal cells, we exposed a mouse model of CRC, as well as wildtype (WT) mice, to these organisms and performed single-cell RNA sequencing (scRNA-seq) on harvested intestinal resections. We utilized an established CRC mouse model that carries a transversion point mutation in one copy of tumor suppressor, *adenomatous polyposis coli* (*Apc*) ($Apc^{Min/+}$). The biallelic loss of *Apc* is detected in 80–90% of CRC patient cohorts and is an initiating event in sporadic CRC [41–43]. This mutation predisposes the mice to intestinal tumors and has been previously used to study the effects of both Fn and ETBF on tumor initiation and progression [8,15,41–44]. Comparing single-cell transcriptional profiles in resections from both WT and $Apc^{Min/+}$ mice afforded the opportunity to disentangle the combined effects of genetics and pathobionts on cellular phenotypes without imposing biases upon which cells these organisms most directly affect.

## Results

### Fn and ETBF alter intestinal cell composition in $Apc^{Min/+}$ and wildtype mice

To determine how CRC pathobionts affect the host intestinal microenvironment, we exposed WT and $Apc^{Min/+}$ mice to Fn or ETBF. Mice received a daily oral gavage of Fn or ETBF at a concentration of $10^8$ colony forming units (CFUs) to expose intestinal cells to the pathobionts [8,15,44,45] (**Fig 1A**). We assessed tumor burden by counting visible tumors upon sacrifice, as described in Kostic *et al.* (2013) [8]. This choice was made due to the presentation of numerous small tumors at 16 weeks. Due to the error in counting tumor volumes for these small tumors, we reasoned that it would be less informative than overall tumor burden. Although Fn and ETBF have been reported to reduce survival rates and increase tumor burdens in $Apc^{Min/+}$ mice, these effects were limited to mice pre-treated with antibiotics [8,45–47]. Although antibiotic exposure is associated with increased CRC risk in humans [48–50], we chose not to pre-treat mice with antibiotics to avoid introducing confounding effects on host tissue either directly or via altered microbiome composition. Of note, this experimental procedure does

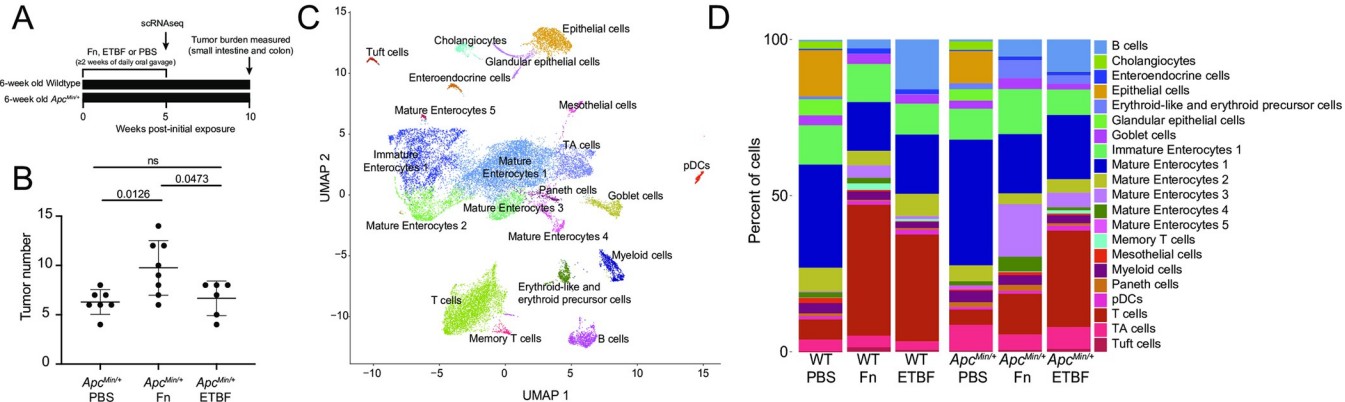

**Fig 1. Exposure to CRC-associated pathobionts results in differences in cellular composition and transcriptional profiles.** (A) Depiction of the experiment. (B) Macroscopic tumor burden in $Apc^{Min/+}$ mice exposed to Fn or ETBF sacrificed at 16 weeks of age (n ≥ 6 $Apc^{Min/+}$ mice). Mice were exposed daily to CRC-associated pathobionts for at least 2 weeks starting at 6-weeks of age. (C) UMAP of transcriptomic profiles of 24,371 cells from all conditions colored according to their annotations. (D) A barplot depicting the composition of cells in each experimental condition.

deviate from established antibiotic-aided colonization methods and may explain why our downstream findings are different from the literature [8,15,44,45]. Nonetheless, we observed greater tumor burden 10-weeks after initial pathobiont exposure in the Fn-exposed $Apc^{Min/+}$ mice (**Fig 1B**), consistent with previous reports [8,51]. We were initially surprised that ETBF administration did not result in increased tumor burden, as it does when ETBF is administered to antibiotic-treated $Apc^{Min/+}$ mice [40,44,45]. ETBF administration, under antibiotic treated conditions, elicits a robust IL-17 driven inflammatory response that mediates the recruitment of myeloid cells and ultimately supports tumor cell growth and proliferation in mice [52]. However, contrary to this pro-tumor phenotype, it is also been shown that ETBF does not increase the mutations-per-megabase and copy number alterations above that observed in $Apc^{Min/+}$ mice that have been pre-treated with antibiotics [47]. Taken together, without antibiotic-mediated colonization and the resultant inflammation, macroscopic tumor induction post-ETBF exposure was likely tempered.

We performed scRNA-seq on intestinal tissue from WT and $Apc^{Min/+}$ mice after oral dosing of Fn or ETBF, or phosphate buffered saline (PBS), as a control. Since Fn and ETBF are enriched in early stages of tumorigenesis (premalignant lesions and adenomas) in CRC patients [53–57], we sacrificed mice at 11-weeks of age corresponding to 5 weeks post-pathobiont exposure or PBS treatment. We transcriptionally profiled 24,371 individual cells, which were clustered into 21 different cellular subsets, using Seurat (version 4.1.1) [58]. Cells were annotated with known cell-type specific marker genes [59,60] and cross-referenced using scMRMA, an automated single-cell annotation algorithm [61] (**Fig 1C**). Cellular compositions across treatment conditions were substantially different, including notable changes across T cells, proliferating enterocyte precursors, and mature enterocytes post-Fn and ETBF exposures (**Fig 1D**).

## Fn and ETBF promote the outgrowth of cancer stem cell-like transit-amplifying cells and cancer-like enterocytes

Transit-amplifying (TA) cells are daughter cells of intestinal stem cells that further differentiate into enterocytes. Due to their high rates of proliferation, they are mutation-prone [62]. Treatment with Fn in co-culture with CRC cell lines has been found to induce the upregulation of stemness associated genes: *CD133*, *CD44*, *Snail1* and *ZEB1* [63,64]. Similarly, ETBF treatment

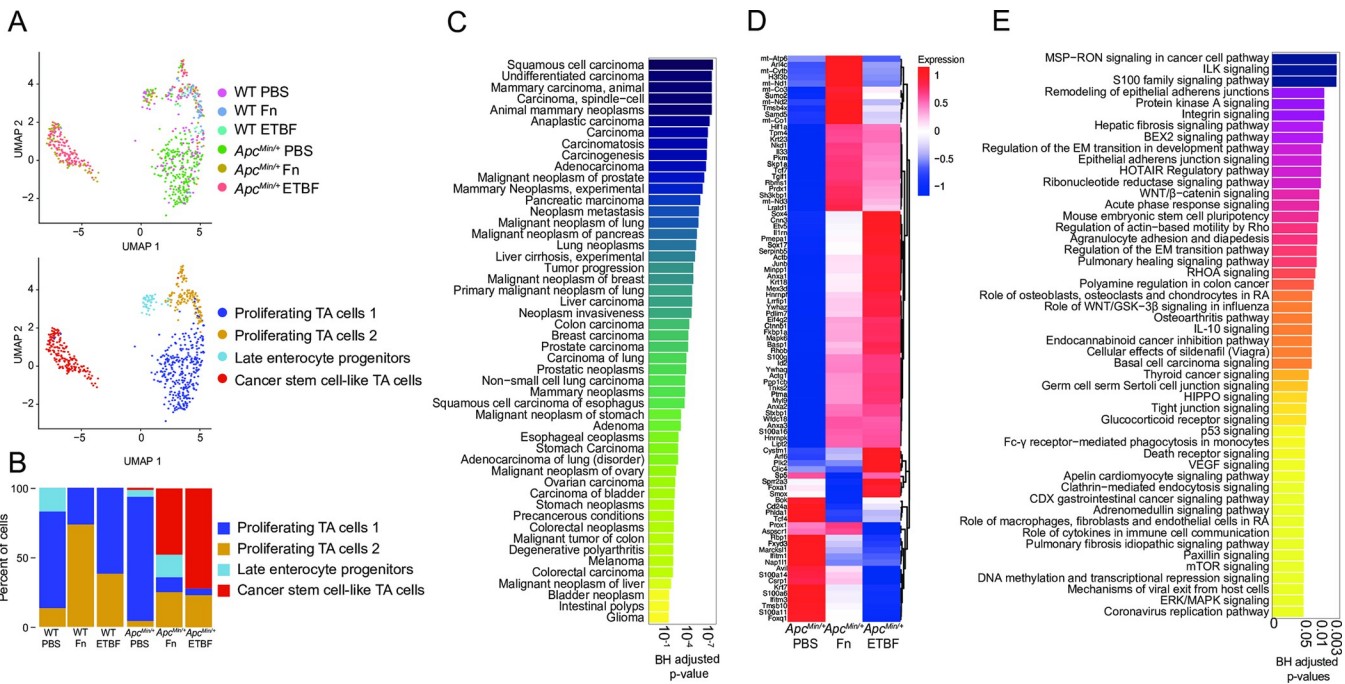

**Fig 2. TA cells from $Apc^{Min/+}$ mice adopt cancer stem-cell like phenotypes after exposure to CRC-associated pathobionts.** (A) UMAP of transcriptomic profiles of TA cells according to experimental condition (top) and subclusters (bottom) (n = 682). (B) A heatmap displaying all 91 upregulated genes for the CSC-like cell cluster (compared to the other TA populations) for each genotype-treatment, (log₂(fold-change) ≥ 0.25 (Wilcox test), Bonferroni-corrected p-value < 0.05, Seurat), plotted as average expression values. (C) A barplot depicting the top 50 IPA Canonical Pathways genesets for the cancer stem cell-like cell population, based on corrected p-values (BH-FDR-corrected p-value < 0.05, IPA). (D) A barplot depicting the top 50 genesets according to DisGeNET (y-axis) for the CSC-like cell population, plotted in descending according to corrected p-values (Fisher exact test, BH-FDR corrected p-values < 0.05, EnrichR). (E) A barplot depicting the percent composition of the cell populations per genotype and treatment.

leads to the increase in stemness in both CRC cell co-cultures and CRC xenograph mouse models, via the upregulation of *JMJD2B*, a histone demethylase [65]. We hypothesized that exposure to Fn and ETBF in $Apc^{Min/+}$ mice would exacerbate neoplastic transformation in these cells accordingly. TA cell transcriptomes sub-clustered into four distinct groups, including one that transcriptionally resembles cancer stem cells (CSCs), based similarities in upregulated genes and pathways between the cells we identified and the known phenotypic profile in the literature [66–69] (**Fig 2A–2D**). Using DEG analysis, we identified 91 genes delineating these CSC-like cells from the other TA cell subpopulations (**Fig 2B**). These include upregulated genes that support intestinal cell survival and proliferation, such as *Foxa1* [70–72], *Sox4* [71,73,74], *Prox1* [75–77], and *Ctnnb1* [78–80] (Fisher exact test p-values < 0.05, BH-FDR corrected p-values < 0.05, EnrichR).

This subpopulation was almost exclusively found in the CRC pathobiont-exposed $Apc^{Min/+}$ mice (**Fig 2E**).

Overall, the CSC-like cells upregulated pro-oncogenic pathways, including integrin and integrin-linked kinase (ILK) signaling, MSP-RON (macrophage-stimulating protein-recepteur d'origine nantais) signaling, and Wnt/β-catenin signaling, among other pathways relating to stem cell pluripotency and the epithelial-mesenchymal transition (EMT) [81–86] (**Fig 2C**) (Fisher exact test p-values < 0.05, BH-FDR corrected p-values < 0.05, Ingenuity Platform Analysis (IPA) canonical pathway analysis, the gene list used as the input for IPA was the result of a comparison (Wilcoxon test) between CSC-like cell clusters and the other three TA cell clusters). There were few significant differentially enriched pathways between these CSC-like

TA cells specific to each pathobiont exposure, although Myc-targeting was comparably elevated in cells derived from $Apc^{Min/+}$ mice exposed to ETBF (**S1 Fig**). As for Fn-exposed CSC-like cell population, fatty acid metabolism was enriched compared to those exposed to ETBF, a finding which is supported by *in vitro* experiments linking this phenotype to enhanced self-renewal (**S1 Fig**) [63]. The top 50 most significant human gene-disease annotations for the DEGs in the CSC-like TA cell population are all cancers, including several related to the colon (**Fig 2D**) (Fisher exact test p-values < 0.05, BH-FDR corrected p-values < 0.05, DisGeNET). These colon-specific gene-disease annotations were unique to the CSC-like TA cells (**S2 Fig**). However, a second cluster of TA cells (proliferating TA cells 2) had similar gene-disease associations to the CSC-like TA cells, albeit different DEGs and enriched pathways. Interestingly, this cluster comprised predominantly cells from wildtype mice exposed to each of the pathobionts (**Figs 2E and S3**). These data suggest that exposure to CRC-associated pathobionts promotes the induction of cancer-stem cell-like cells within the $Apc^{Min/+}$ mice that possess transcriptomic hallmarks of human cancer stem cells.

Mature enterocytes, derived from TA cells, are directly exposed to the microbiome and make up the vast majority of the cells within CRC tumors [68,87]. Both Fn and ETBF treatment increases tumor burden due to the outgrowth of transformed enterocytes in certain mouse models and drive rapid proliferation of epithelial cell lines in co-culture experiments [8,16,26,31,88,89]. Within the mature enterocyte cell population, we performed unsupervised clustering on cellular transcriptional profiles, resulting in four groups (**Fig 3A**). One group was noticeably enriched for cells derived from $Apc^{Min/+}$ mice exposed to Fn and ETBF and displayed a unique cancer-associated profile (**Fig 3B and S4**). Within this subset, 693 genes are differentially upregulated compared to the other three enterocyte sub-clusters, including the Wnt signaling mediator *Ctnnb1*, canonical cancer markers *STAT3* and *HIF1α*, and *Klf3*, *Klf4*, *Klf5* and *Klf6*, all of which exhibit tumor suppressive properties in many cancers, including CRC [80,90–92] (**Figs 3C and S4**). When compared to all other mature enterocyte sub-populations, the DEGs for this subset were enriched for genesets involved in PI3K/AKT/mTOR signaling, p53 signaling and apoptotic pathways (**Fig 3D**) (Fisher exact test p-values < 0.05, BH-FDR corrected p-values < 0.05, EnrichR). Analysis using the IPA platform was consistent with DisGeNET, showing a significant enrichment of disease and functional annotations associated with tumorigenesis (**S4 Fig**). Overall, these data suggest that this mature enterocyte population from pathobiont-exposed $Apc^{Min/+}$ mice adopts a cancer-like phenotype, like that observed in TA cells from the same mice.

Together, these results support a model in which these pathobionts can influence cancer-associated signaling cascades, CRC initiation via CSC-like cell population induction and CRC progression by cancer-like enterocyte enrichment within the context of $Apc^{Min/+}$ mouse model. Supporting our work, a recent study investigating the interplay between Fn and human CRC tumors found that epithelial cell population with a high Fn burden upregulated Myc, mTORC1 and PI3K-AKT-mTOR signaling pathways. This important finding suggests that the enrichment of cell growth and proliferation signaling programs are a specific deleterious outcome elicited by Fn and in our study, ETBF as well [7].

## Pathobionts elicit similar effects in both-specific effects on cytotoxic T cells are abrogated in $Apc^{Min/+}$ mice

T cells are critical for tumor immunosurveillance [93,94]. However, the colorectal tumor microenvironment drives T cells, including potent anti-cancer CD8[+] cytotoxic T lymphocytes (CTLs), towards immunosuppressive, senescent, and exhaustive states [95–97]. In addition, CRC pathobionts Fn and ETBF exhibit profound T cell modulatory effects. In previous studies

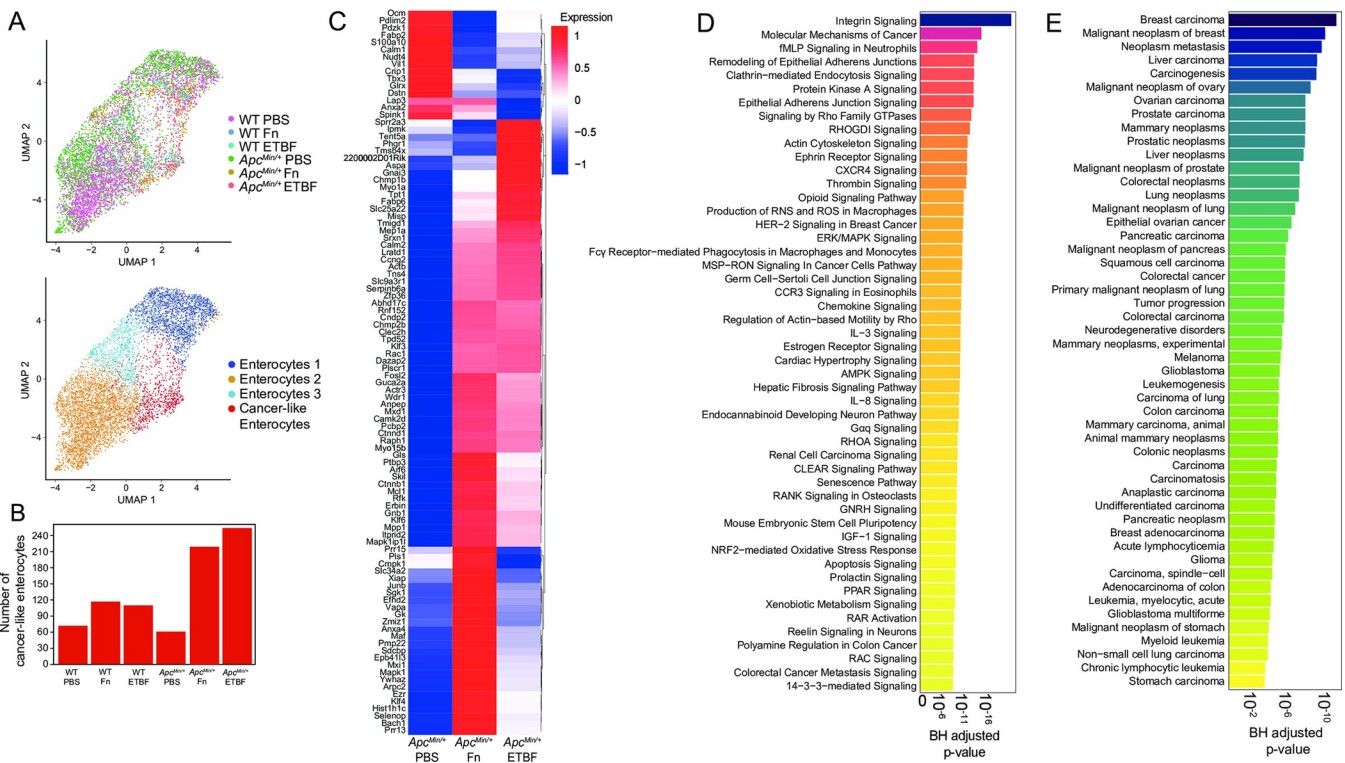

**Fig 3. Mature enterocytes from pathobiont-treated $Apc^{Min/+}$ mice display cancer-like phenotypes.** (A) UMAP of transcriptomic profiles of 6,719 enterocyte populations colored by experimental condition(top) and by sub-clusters (bottom). (B) A barplot displaying the number of cells within each sub-cluster, according to experimental condition. (C) A clustered heatmap displaying the top 100 upregulated genes (log2(fold change) ≥ 0.25 (Wilcoxon test, BH-FDR corrected p-values < 0.05, Seurat), plotted as average expression values (Seurat) for the cancer-like enterocytes compared to all other enterocyte populations. (D)A barplot depicting the top 50 IPA Canonical Pathways genesets (y-axis) based on corrected p-values (Fisher exact test, BH-FDR corrected p-values < 0.05, IPA) for the cancer-like enterocytes. (E) A barplot depicting the top 50 genesets according to DisGeNET for the cancer-like enterocyte population, plotted in descending according to corrected p-values (Fisher exact test, BH-FDR corrected p-values < 0.05, EnrichR).

using $Apc^{Min/+}$ mice, ETBF exposure led to enhanced T cell differentiation skewing towards Th17 cells and away from CTLs, albeit this effect was indirect, mediated through the recruitment and activation of myeloid derived-suppressor cells (MDSCs) [40,98]. Similarly, Fn triggers the expansion of MDSCs in $Apc^{Min/+}$ mice, although without any effect on T cell populations [8]. However, in humans, Fn abundance within the tumor is inversely correlated with tumor-specific T cell abundances [99], and in cell culture, Fn directly binds human T cells and inhibits their function, potentially via interactions between TIGIT and Fn adhesin, Fap2 [24,100]. Nevertheless, we did not observe specific changes involving TIGIT engagement because mouse TIGIT does not bind to Fap2 [24]. To define the T cell subsets in our single-cell dataset, we characterized 3,101 T cells. The cells were partitioned using marker genes, yielding 4 subclusters: CTLs, γδ T cells, T regulatory cells, and mucosal-associated invariant T cells (**Fig 4A**). We focused on characterizing the CTL population, based on previous observations, and because they possess the cytotoxic function essential to the ablation of tumor growth. We also investigated whether microbe-specific transcriptional changes occurred in the myeloid cell compartments and although the myeloid cell counts were considerably low, proinflammatory macrophages derived from the Fn-treated $Apc^{Min/+}$ mouse were enriched for positive regulation of SMAD signaling and epithelial-to-mesenchymal transition compared with those from the WT mouse, though pathways did not pass the Bonferroni-correction threshold. (**S5 Fig**) (Fisher's exact test p-values < 0.05, Bonferroni-corrected p-values < 0.05, EnrichR). The

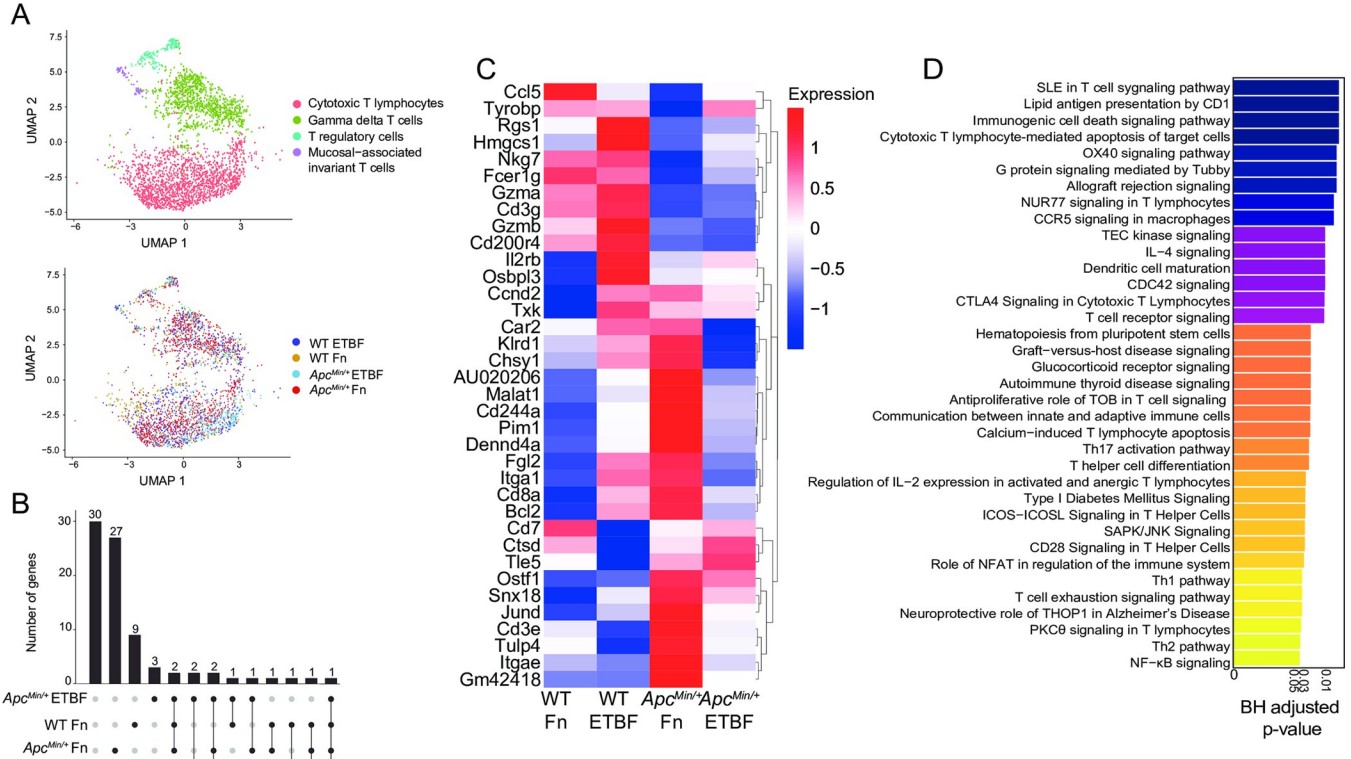

**Fig 4. Pathobionts elicit similar effects in both-specific effects on cytotoxic T cells are abrogated in Apc^Min/+ mice.** (A) UMAP of transcriptomic profiles of 3,101 T cell populations colored by sub-cluster (top) and by experimental condition (bottom). (B) Upset plot depicting the differentially expressed genes that each CTL population (log$_2$(fold-change) ≥ 0.25, Wilcoxon test, BH-FDR-corrected p-value < 0.05) based on sample, the set size is the total number of genes expressed and the intersection size the number of genes that are shared by dataset, an individual sample alone indicating that the genes are only expressed by the cells in that dataset and lines representing shared genes. (C) A heatmap displaying the top 36 upregulated genes (log$_2$(fold-change) ≥ 0.25, Wilcoxon test, BH-FDR-corrected p-value < 0.05), plotted as average expression values (Seurat) for the cytotoxic T lymphocytes across each dataset. (D) Barplot depicting the top 36 IPA Canonical Pathways genesets (y-axis) based on corrected p-values (Fisher exact test, BH-FDR corrected p-values < 0.05, IPA). The gene list used as input for canonical pathway analysis were the genes upregulated by ETBF-exposed WT CTLs, when compared to ETBF-exposed Apc^Min/+ CTLs.

numbers of CTLs isolated from the PBS control animals were also low and were therefore removed from downstream analyses. Of the genes that define the CTL cluster, made up of cells from pathobiont-exposed mice, we observed that genes central to CTL function, including the cytolytic granule constituents, *Gzma* and *Gzmb* [101,102], and to a lesser extent *Cxcr6* [103–105], a chemokine receptor, are upregulated in the WT pathobiont-exposed mice, but not the *Apc^Min/+* pathobiont-exposed mice (**Fig 4B and 4C**). These results suggest that the *Apc^Min/+* background, possibly due to tumor-mediated immunosuppression, can mollify cytolytic CTL responses that are observed in wild-type post-pathobiont exposed counterparts.

To better understand how the *Apc^Min/+* model affects CTLs post-ETBF exposure, we compared the transcriptional profiles from ETBF-exposed *Apc^Min/+* with ETBF-exposed WT mice. We found WT ETBF-exposed CTLs upregulated genesets involved in cytotoxic T lymphocyte–mediated apoptosis of target cells, T cell receptor signaling and OX40 signaling pathway [106–108], suggesting that ETBF treatment under normal conditions elicits a robust CTL response, and that this is suppressed in the *Apc^Min/+* mice (Fisher's exact test p-values < 0.05, Bonferroni-corrected p-values < 0.05, IPA canonical pathway analysis) (**Fig 4D**). These results further support a model where CRC pathobionts induce T-cell dependent immunogenicity that is largely abrogated when tumors are present.

## Discussion

Recent cancer pathophysiology studies have shown that the gut microbiota can play a significant role in tumor initiation, progression, or both [109–112]. Within CRC patients' gut microbiomes, organisms such as Fn and ETBF act as pathobionts, because of their ability to induce host inflammation, DNA damage, and cell proliferation [109–111]. These bacteria are thought to initiate the formation of carcinogenic bacterial biofilms and antagonize host immunity by tempering anti-tumor immunity [14,15,24]. Despite a growing body of evidence supporting the role of bacteria in CRC tumor burden and patient survival [109–111], much of the work uncovering the mechanisms underpinning this phenomena have been restricted to experiments using cell culture or on specific cell types isolated from mouse models.

The scRNA-seq data presented here suggests that there are cell-specific and pathobiont-specific effects evident in immune and epithelial tissue. Our analysis reveals that Fn and ETBF can provoke a CSC-like transcriptional profile in TA cells. These CSC-like TA cells bridge pathophysiological observations with specific cellular responses, including, but not limited to, known stemness genes. Moreover, mature enterocytes, which appear to be susceptible to neoplastic transformation, are an emergent feature of $Apc^{Min/+}$ intestinal cell profiles post-Fn and ETBF exposure. CTLs, on the other hand, displayed transcriptiomes evident of reduced cytotoxic capacity in pathobiont-exposed $Apc^{Min/+}$ mice, when compared to their pathobiont-exposed wildtype counterparts. By directly comparing Fn- and ETBF-exposed mice, we observed consistent features invoked by both pathobionts in TA, enterocyte and CTL populations. These results suggest that pathobiont exposure can foster an environment conducive to the outgrowth of tumorigenic intestinal cell populations.

The effects on TA cells, enterocytes and cytotoxic T lymphocytes that we observe were each affected by the underlying genetic background of the CRC mouse model we used. The $Apc^{Min/+}$ mouse model recapitulates a relevant mutation in human CRC (80–90% of all sporadic CRC cases) and is therefore the most widely utilized mouse model for CRC. However, there are some notable differences between this model's pathophysiology versus that which is observed in humans. For instance, the primary site of tumorigenesis in the $Apc^{Min/+}$ mouse is the small intestine, rather than the colon [113]. Examining the effects of CRC-associated pathobionts in additional mouse models of CRC, including those that exhibit greater colonic tumor burden (*e.g.* mice carrying inducible mutations in *Apc*, *Kras*, and *p53* specific to the colon, such as those driven by Villin or Cdx2) [114] could further enhance our understanding of colon-specific tumorigenesis mediated by Fn and ETBF. Notwithstanding these alternatives, the $Apc^{Min/+}$ model affords the ability to elucidate microbe-specific transcriptional responses in a system free of numerous cancer drivers and in a model within which these organisms have shown to affect tumorigenesis.

This study demonstrates the effects of repeat exposure to CRC pathobionts. There are several limitations of our experimental design. First, we did not use antibiotics nor germ-free mice, as we wanted to maintain the native murine microbiome. This came with the caveat that without antibiotics, Fn and ETBF colonization is not robust. Rather, our results highlight the cellular effects of short-term repeat exposure on intestinal tissue. These results support the hit-and-run carcinogenesis model [115–118], whereby CRC pathobionts exposure is transient but the pro-tumor effects elicited pathobionts manifest by experimental endpoints. Additionally, we were interested in providing a detailed single-cell characterization of both epithelial subtypes and immune cells from both small intestine and colon. For that reason, we pooled and sequenced cells from both anatomical sites. By doing this, we were able to capture epithelial cell heterogeneity, including the detection and characterization of cancer stem cell-like transit-amplifying cells and cancer-like enterocytes. While this method of single-cell preparation

reduced our ability to capture immune cells and other lower abundance cell types such as Paneth and enteroendocrine cells in particular, we avoided examining transcriptional changes induced by cell enrichment methods [119].

Transient exposure, rather than colonization, may have tempered the pro-tumorigenic effects of ETBF (Fig 1B), and possibly Fn, via niche exclusion and/or colonization resistance [120–122]. Moreover, transient exposure and lack of antibiotic use could limit the pathobiont's access to many of the cell populations traditionally associated with their pathogenic inflammatory etiology such T cells and macrophages, which largely are in the lamina propria, and spatial distance from direct interactions with Fn and ETBF, and their pathogen-associated molecular patterns [123,124]. Nonetheless, we still find that transient exposure to Fn and ETBF in the $Apc^{Min/+}$ model triggers transcriptional programs that support the outgrowth of CSC-like cells and cancer-like enterocytes. Similar short-term exposures to ETBF induces robust cytotoxic T cell responses in wildtype mice. Taken together, this suggests that Fn and ETBF pro-tumor effects could be more robust than previous thought.

Fn and ETBF are known for their ability to trigger distinct tumor promoting mechanisms. Fn adhesin FadA modulates aberrant Wnt signaling via E-cadherin and β-catenin in enterocytes [26,27]. ETBF possesses a DNA damaging toxin, Bft, and induces Myc signaling in enterocytes and an inflammatory immune cascade largely mediated by Th17 cells and IL-17 [34,35,38]. One of our study's important findings is that Fn and ETBF, despite their unique tumorigenic proclivities, mostly overlap mechanistically as evidenced by the similar cancer-associated transcriptional programs evoked in enterocyte and enterocyte pre-cursors. This suggests that both organisms have common CRC initiating and/or supporting characteristics that affect similar cell types. These findings were enabled by the significant number of enterocytes sequenced across our murine intestinal samples. Herein lies a key shortcoming as well, which does not represent common biology. By probing thousands of enterocytes, other rarer cell types were found in smaller numbers. For this reason, comparative analyses between Fn and ETBF treatments across almost all other cell types, including across both $Apc^{Min/+}$ and wild-type mice, were underpowered, and we could not delineate statistically significant differences (BH corrected p-value < 0.05). Nevertheless, our findings still represent an important step in delineating enterocyte and TA cell-specific transcriptomic changes post CRC pathobiont exposure and warrants future investigations delving into larger swath of intestinal cells in depth.

Although Fn and ETBF are perhaps the most well-known CRC-associated pathobionts, a fuller picture of CRC initiation and progression likely involves other key microbial players. For example, $pks^+$ E. coli is an E. coli strain that produces colibactin, a genotoxin that cause double strand breaks in the intestinal cells' DNA also has the ability to transform cells [125–127]. The development of polymicrobial biofilms is another emergent feature of CRC. Biofilms are significantly enriched in right sided colon adenomas (precancerous lesion) versus adjacent healthy tissue and have been causally linked with CRC in mouse models [14,15,128]. Additionally, other oral pathobionts beyond Fn, such as *Parvimonas micra*, *Peptostreptococcus stomatis*, *Peptostreptococcus anaerobius* and *Gemella morbillorum*, are commonly enriched in patients with CRC [111,129,130]. Experimentally, *P. anaerobius* and *P. micra* having been shown to play a causal role in oncogenesis in azoxymethane and $Apc^{Min/+}$ mouse models, respectively [131,132]. Pertaining to these organisms, major questions in the field remained about how these oral microbes, in concert with gut pathobionts, seed biofilms and, if so, whether the biofilms promote tumorigenesis in the colon [126,133–136]. Performing similarly designed scRNA-seq experiments using additional organisms and eventually consortia will likely be invaluable in delineating the modulatory effects gut bacteria have on CRC tumor initiation and development.

Tumor-specific microbiomes, biofilm formation, and microbiome dysbiosis are all implicated in CRC progression. Using scRNA-seq, we were able to reconstruct cell type-specific effects that occur post-pathobiont exposure. However, recently developed approaches that enable combined host transcriptomics with microbiome species mapping [137,138] will provide additional spatial contextualization, directly associating specific gut microbiota with cell-specific transcriptional changes occurring within the tumor microenvironment. Studying the effects of Fn, ETBF and other pathobionts *in vivo*, using unbiased approaches like these offer the promise of identifying marker genes that may be used to enhance cancer diagnostics and therapeutics.

## Materials and methods

### Ethical considerations

This study conformed to the National Institutes of Health guidelines on the care and use of laboratory animals. Mouse studies were performed following procedures approved by the Institutional Animal Care and Use Committee at Cornell University (Protocol ID #2016– 0088). Mice were monitored daily by staff at the Center for Animal Resources and Education (CARE) and sacrificed either at the end of the study or at ethical endpoints: any indication of poor health including but not limited to the following: decreased activity, dehydration, abnormal fur changes, ataxia, and/or excess weight loss (20% loss of total body weight). Any distress would result in an notification and mice were provided with heat pads and wet chow. Additional water was provided in closer proximity to the ground. Any type of leak in the water was remedied the same day, with dry caging and bedding provided. All researchers handling mice received training through CARE. Due to the nature of the mouse model used in these experiments, in which numerous small (visible) tumors are formed, we chose to assess tumor burden rather than tumor volume, which is more commonly used for xenograft tumor experiments.

### Bacterial strains and culturing

*Fusobacterium nucleatum* subsp. nucleatum strain VPI 4355 [1612A] (ATCC 25586) was purchased from American Type Culture Collection (ATCC). *Bacteroides fragilis* (Veillon and Zuber) Castellani and Chalmers strain 2-078382-3 (ATCC 43858) (ETBF) was purchased from American Type Culture Collection (ATCC). Fn and ETBF were grown anaerobically at 37˚C on Bacto™ Brain Heart Infusion Broth (BD, Sparks, MD) supplemented with 0.01% Hemin in 1M NaOH, 0.1% Resazurin (25 mg/100ml distilled water), 10% NaHCO3 in distilled water, and agar if bacteria were plated. Bacteria were grown overnight and diluted to $10^8$ colony forming units (CFU), the amount needed for oral gavage.

### Mice

All mice (C57BL/6-$Apc^{Min/+}$/J and C57BL/6-Wild type) were maintained at the barrier mouse facility at Weill Hall at Cornell University. $Apc^{Min/+}$ and wild-type mice were initially ordered from Jackson Laboratory and then bred in the barrier facility. The $Apc^{Min/+}$ mice used in these experiments have a chemically induced transversion point mutation (a T to an A) at nucleotide 2549. This results in a stop codon at codon 850, truncating the APC protein. Both male and female mice were used in all experiments. Experimental and breeding mice were provided with *ad libitum* access to autoclaved water and rodent chow (autoclavable Teklad global 14% protein rodent maintenance diet #2014-S; Envigo). To avoid cage effects on the microbiota, mice were housed individually at the time of initial Fn, ETBF or PBS exposure. One mouse per condition (6 mice total) were used for the scRNA-seq experiments, whereas ten mice started

the tumor burden study. To monitor for infectious agents such as helminths, sentinel mice were used during the duration of the experiment in the mouse facility to ensure that results following perturbation with Fn and ETBF were a result of specific bacteria and not confounding agents. Every week, food intake and animal weight were recorded, and mice were placed in clean cages with freshly autoclaved chow and water weekly. Food intake and weight was recorded to ensure that mouse tumor burden did not violate ethical standards. Mice were handled under inside a biosafety cabinet with frequent glove changes and disinfection between mice during stool collection and monitoring of body weight. Stool was collected weekly throughout the course of all experiments. Bacterial oral gavage experiments were performed every day for a period of at least 14 consecutive days for ETBF, and up 35 days for Fn [8,25,45], beginning at 6 weeks of age. Bacteria were fed at a concentration of $10^8$ CFU per day. Sham treatment consisted of sterile $Ca^{2+}$ and $Mg^{2+}$ free phosphate buffered saline gavaged daily for the entirety of the experiment. Single-cell RNA experiments concluded when the mice were 11 weeks old and tumor burden experiments concluded when mice were 16 weeks old. Several mice in the tumor burden study reached humane endpoints prior to 16 weeks and were not used in assessing tumor burden (3, 2 and 4 mice in the PBS, Fn and ETBF $Apc^{Min/+}$ groups respectively). Mice of both sexes were used for all experiments and were monitored daily. Mice were sacrificed using 5 minutes of $CO_2$ asphyxiation either at the end of the study or when they reached humane endpoints (see above).

## Tumor burden enumeration

For tumor enumeration, $Apc^{Min/+}$ mice were euthanized at 16 weeks of age, and colons and small intestines were excised. Macroscopic (visible) tumors were counted from both anatomical sites, as established previously in Kostic *et al.* (2013) [8]. This model differs from xenograft experiments in which tumor burden is often measured. Due to the numerous small tumors formed in the $Apc^{Min/+}$ mouse model, tumor volume could not be calculated accurately; we therefore relied on tumor enumeration. Tumor counts were plotted using Prism (version 8.2.1). For statistical analysis, Mann-Whitney two-tailed tests were used to compare treatment groups using Prism. Each groups had an n $\geq$ 6 mice.

## Single cell dissociation from fresh mouse colons and small intestines

This protocol was adapted from Haber et al 2017 [59]. To generated single-cell suspensions, $Apc^{Min/+}$ and wild type mice were euthanized at 11 weeks of age, colons and small intestines were excised, rinsed with ice cold sterile 1X $Ca^{2+}$ and $Mg^{2+}$ free PBS (Gibco, 14190144) and flushed of fecal contents using a blunt 1.5-inch 22G needle filled with ice cold sterile 1X $Ca^{2+}$ and $Mg^{2+}$ free PBS (Gibco, 14190144). The tissue was opened longitudinally and sliced into small fragments roughly 1 cm in length. The tissue was incubated in RPMI supplemented with L-glutamine (Corning, 45000–396), 1 mM EDTA (Neta Scientific, QB-A611-E177-10), and 10% FBS (Avantor, 97068–085) for 90 minutes, shaking every 30 minutes. The tissue was then incubated at 37°C for 15 minutes and continuously shaken. The supernatant was passed through a 100 µm cell strainer and held on ice until loading the cells on 10X Chromium. The remaining tissue was resuspended in RPMI (Corning, 45000–396) supplemented with 20% FBS (Avantor, 97068–085), 0.1 mg/ml DNase I (Thermo Scientific, 90083), and 0.5 mg/ml collagenase A (Millipore Sigma, 10103586001) and incubated at 37°C on a shaker for 30 minutes. The tissue was then gently mechanically dissociated using a rubber plunger of a syringe. The tissue and the dissociated contents were passed through a 100 µm cell strainer. The single cell suspension was then pelleted via centrifugation (400 x g for 10 minutes at 4°C). The cell suspension was resuspended in 1X $Ca^{2+}$ and $Mg^{2+}$ free PBS (Gibco, 14190144) containing 0.04%

weight/volume BSA (VWR, 97061–420) and combined with earlier collected fraction and placed on ice. Sample viability was determined before loading the cells on 10X Chromium using the Countess II Automated Cell Counter (ThermoFisher). The desired number of transcriptomes from viable cells for each sample was 5000–6000 cells per sample.

## Single-cell RNA sequencing library preparation

5000–6000 viable ($\geq$ 70% alive) cells per sample (from colon and small intestine tissues) were targeted on the 10X Genomics Controller using one lane per mouse/sample for Gel Beads in Emulsion (GEM). Cells from the small intestine and colon were pooled together before GEM creation. Briefly, cells were separated into GEMs along with beads coated in oligos that capture mRNAs using a poly-dT sequences. This was followed by cell lysis and barcoded reverse transcription of mRNA, followed by amplification, and enzymatic fragmentation and 5′ adaptor and sample index attachment. Single-cell libraries were generated using the Chromium Next GEM Single Cell 3' Library Construction V3 Kit (10X Genomics) and were then sequenced on an Illumina NextSeq 2000 run with the 100 bp P2 kit for all samples. Sequencing data were aligned to the mouse reference, mm10 (Ensembl 84) reference genome using the Cell Ranger 5.0.1 pipeline (10X Genomics).

## Single-cell RNAseq data processing and visualization

The output of Cell Ranger is a cell-by-gene unique molecular identifier (UMI) expression matrix for each sample. The expression matrices for each sample are loaded into the Seurat R package (Seurat version 4.1.1, R version 4.1.0 and 4.2.0). The standard Seurat dataset processing workflow was followed. In brief, cells with less than 200 genes, more than 2,500 genes, and more than 35% mitochondrial genes are filtered out. After filtering, the remaining cells were normalized by the total expression, multiplied by the default scale factor (10,000), and log transformed. We then used default Seurat functions to identify highly variable genes with one parameter modification. FindVariableFeatures' nfeature parameter was set to 3,000 instead of 2,000 (default). Next, we scaled the data to regress out variation from mitochondrial genes. We performed principal component analysis (PCA) on the scaled data with variable genes. The top 20 principal components were used for downstream analysis, including dimensionality reduction steps including clustering cells to identify cell populations (clusters). We implemented Uniform Manifold Approximation and Projection for dimensional reduction using the top 20 PCs and visualized.

## Marker-gene identification and cell-type annotation

To define cell types for each cluster, we used Seurat's FindAllMarkers with the following parameters: a minimum percent expression value of 25%, log$_2$fold change threshold of 0.25 and a corrected p-value < 0.05 (Bonferroni correction). We looked only at transcripts that were upregulated. We analyzed canonical markers and assigned cell annotations accordingly. We cross-referenced our cell type annotations with gene lists defined in Haber et al. [59] and Moor et al. [60] We cross-reference the cell type assignments with a single cell annotation algorithm, scMRMA in R as well [61].

## Reclustering, visualization, and analysis of transit-amplifying cells, mature enterocyte (1) and T cell populations

We used the 682 TA cells, 6,719 mature enterocytes (1), and 3,101 T cells and re-clustered them using Seurat. Marker genes for each subclusters were identified using a minimum

percent expression value of 25%, log$_2$fold change threshold of 0.25 and a corrected p-value < 0.05 (Bonferroni correction) in Seurat. Cell types were assigned based on the expression of these marker genes. Cell clusters expressing marker genes from multiple unrelated cell types (doublets) were removed from analysis. All sub-clustering analysis was carried out with 20 principal components and similar resolution parameters; TA cells and T cells were analyzed with a resolution of 0.4 and mature enterocytes (1) with a resolution of 0.3 in Seurat. The marker gene list used to classify cell subtypes can be found in S1 Table. Cell populations were visualized using Uniform Manifold Approximation and Projection in Seurat. Cell were enumerated, whether as percent of sample or absolute count, using the dittoSeq's (version 1.8.1) bar plot visualization function.

## Differential gene expression and geneset enrichment analysis

Differentially gene expression was carried out using Seurat's FindAllMarkers and FindMarkers functions with the following cutoffs: log$_2$(fold change) $\geq$ 0.25 (Wilcox test), corrected p-value < 0.05 (Bonferroni correction) and a minimum percent expression value of either the default, 10%, or 25% for certain other analyses. For these analyses, only upregulated genes were used. We visualized DEGs using the Seurat's DoHeatmap and dittoHeatmap (dittoSeq) for heatmaps, dittoPlot(dittoSeq) for violin plots and UpSetR (version 1.4.0) for upset plots. For statistics associated with violin plots (S4 Fig), we performed a two-sample Wilcoxon test, comparing each normal enterocyte cluster against the cancer-like enterocyte cluster using the stat_compare_means function in ggpubr (version 0.5.0). For gene set enrichment analysis, the gene list used as input were generated as detailed above using FindMarkers (Seurat). A suite of tools and databases were implement for these analysis and are as follows: Ingenuity Pathway Analysis (IPA, Qiagen) including canonical pathway and disease and function analysis, DisGe-NET (version 7.0) via Enrichr [139,140], and MSigDB Hallmarks 2020 via EnrichR [140].

## Supporting information

**S1 Fig. CSC-like TA cells from Fn- and ETBF-exposed *Apc*$^{Min/+}$ mice differed in key pathways.** (A) Top 20 differentially enriched pathways (MSigDB Hallmarks 2020) represented in the transcriptomes of cells from CSC-like TA cells from the Fn-exposed *Apc*$^{Min/+}$ mouse as compared to the PBS-treated *Apc*$^{Min/+}$ mouse. (n = 175 cells, Fisher exact test, BH-FDR-corrected p-values < 0.05, EnrichR) (B) Top 3 differentially enriched pathways (MSigDB Hallmarks 2020) represented in the transcriptomes of cells from CSC-like TA cells from the ETBF-exposed *Apc*$^{Min/+}$ mouse as compared to the PBS-treated *Apc*$^{Min/+}$ mouse. (n = 175 cells, Fisher exact test, BH-FDR-corrected p-values < 0.05, EnrichR). S1 Fig complements Fig 2.
(TIF)

**S2 Fig. Cancer-specific gene-disease associations with DEGs identified in TA cells were specific to those from pathobiont-exposed *Apc*$^{Min/+}$ mice.** (A) A barplot depicting the top 50 genesets according to DisGeNET (y-axis) for the proliferating TA cells (1), plotted in descending according to corrected p-values (x-axis, Fisher exact test, BH-FDR corrected p-values < 0.05, EnrichR). (B) A barplot depicting the top 50 genesets according to DisGeNET (y-axis) for the proliferating TA cells (2), plotted in descending according to corrected p-values (Fisher exact test, BH-FDR-corrected p-values < 0.05, EnrichR). (C) A barplot depicting the top 14 genesets according to DisGeNET (y-axis) for the late enterocyte progenitors, plotted in descending according to corrected p-values (Fisher exact test, BH-FDR corrected p-values < 0.05, EnrichR). S2 Fig complements Fig 2.
(TIF)

**S3 Fig. Proliferating TA cells 2, similar to CSC-like TA cells in notable disease associations, diverge at the gene and pathway levels.** (A) The TA cells depicted here are the 4 subclusters of the complete TA cell population and are an aggregate from all mouse samples ($Apc^{Min/+}$ mice treated with PBS, Fn or ETBF and wild type mice treated with PBS, Fn or ETBF. A heatmap displaying the top 20 upregulated genes for each TA cluster, $\log_2$(fold-change) $\geq 0.25$ (Wilcox test), corrected p-value $< 0.05$ (Bonferroni correction), Seurat), plotted as average expression values (Seurat). (B) Differentially enriched pathways represented in the transcriptomes of proliferating TA cells 2 compared with other TA cell populations. Barplot depicting the top 10 genesets according to the Molecular Signatures Database Hallmark 2020 (MSigDB Hallmarks 2020) for the cancer-like cell population, plotted in descending according to corrected p-values (Fisher exact test, BH-FDR corrected p-values $< 0.05$, EnrichR). S3 Fig complements Fig 2 and S2 Fig.
(TIF)

**S4 Fig. Transcriptome profiles of cancer-like enterocytes were enriched in cancer-like genes and pathways.** (A) Violin plots displaying selected CRC-associated genes and their expression levels across 4 enterocyte clusters ($\log_2$(fold-change) $\geq 0.25$, Wilcoxon test, Bonferroni-corrected p-value $< 0.05$). (B) Barplot depicting the top 50 IPA Diseases and Functions annotations based on corrected p-values (Fisher exact test, BH-FDR corrected p-values $< 0.05$,) for the cancer-like enterocyte subpopulation. Statistical comparisons were performed using a pairwise Wilcoxon test (* = $p \leq 0.05$, ** = $p \leq 0.01$, *** = $p \leq 0.001$, **** = $p \leq 0.0001$), comparing the cancer-like enterocyte population to all other mature enterocyte clusters (see S4 Fig). S4 Fig complements Fig 3.
(TIF)

**S5 Fig. Proinflammatory macrophages derived from the Fn-exposed $Apc^{Min/+}$ mouse upregulate pathways associated with TGF-β/SMAD signaling and epithelial-to-mesenchymal transition.** (A) A heatmap displaying the top 50 upregulated genes defining the proinflammatory macrophage population compared across each dataset ($\log_2$(fold-change) $\geq 0.25$, Wilcoxon Rank Sum test, p-value $< 0.05$ (unadjusted), Seurat), plotted as average expression values. (B) Barplot depicting the top 50 enriched genesets according to the Gene Ontology Biological Processes 2021 (GOBP21) for proinflammatory macrophages derived from Fn-exposed $Apc^{Min/+}$ mice when compared to PBS control $Apc^{Min/+}$ mice, plotted in descending according to p-values (Fisher exact test p-values $< 0.05$, unadjusted, EnrichR). (C) Barplot depicting the top 50 enriched genesets according to the Gene Ontology Biological Processes 2021 (GOBP21) for proinflammatory macrophages derived from Fn-exposed $Apc^{Min/+}$ mice when compared to ETBF exposed $Apc^{Min/+}$ mice, plotted in descending according to p-values (Fisher exact test p-values $< 0.05$, unadjusted, EnrichR).
(TIF)

**S1 Table. Genes used to classify single cells.** The top 10 differentially expressed genes per cluster listed were used to classify single-cells into cell types. Marker genes were defined using the FindAllMarkers function in Seurat ($\log_2$(fold-change) $\geq 0.25$ (Wilcox test), corrected p-value $< 0.05$ (Bonferroni correction)). The top 10 marker genes were included for each cluster.
(DOCX)

## Acknowledgments

We thank the de Vlaminck lab for helpful conversations and feedback.

## Author Contributions

**Conceptualization:** Josh Jones, Ilana L. Brito.

**Data curation:** Josh Jones.

**Formal analysis:** Josh Jones, Rahul R. Nath.

**Funding acquisition:** Ilana L. Brito.

**Investigation:** Josh Jones, Qiaojuan Shi, Ilana L. Brito.

**Methodology:** Josh Jones, Qiaojuan Shi, Ilana L. Brito.

**Project administration:** Qiaojuan Shi, Ilana L. Brito.

**Supervision:** Qiaojuan Shi, Ilana L. Brito.

**Validation:** Josh Jones.

**Visualization:** Josh Jones, Rahul R. Nath.

**Writing – original draft:** Josh Jones, Ilana L. Brito.

**Writing – review & editing:** Josh Jones, Qiaojuan Shi, Ilana L. Brito.

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
