## [Decision Letter · Decision Letter 0]

10 Aug 2023

PONE-D-23-17692Keystone pathobionts associated with colorectal cancer promote oncogenic reprogramingPLOS ONE

Dear Dr. Brito,

Thank you for submitting your manuscript to PLOS ONE. After careful consideration, we feel that it has merit but does not fully meet PLOS ONE’s publication criteria as it currently stands. Therefore, we invite you to submit a revised version of the manuscript that addresses the points raised during the review process.

We look forward to receiving your revised manuscript.

Kind regards,

Syed M. Faisal, Ph.D.

Academic Editor

PLOS ONE

Journal Requirements:

4. Please note that funding information should not appear in any section or other areas of your manuscript. We will only publish funding information present in the Funding Statement section of the online submission form. Please remove any funding-related text from the manuscript.

7. We note that you have included the phrase “data not shown” in your manuscript. Unfortunately, this does not meet our data sharing requirements. PLOS does not permit references to inaccessible data. We require that authors provide all relevant data within the paper, Supporting Information files, or in an acceptable, public repository. Please add a citation to support this phrase or upload the data that corresponds with these findings to a stable repository (such as Figshare or Dryad) and provide and URLs, DOIs, or accession numbers that may be used to access these data. Or, if the data are not a core part of the research being presented in your study, we ask that you remove the phrase that refers to these data.

Additional Editor Comments:

The manuscript by Josh Jones et al. examines Fn and ETBF's impact on colorectal cancer (CRC) through single-cell RNA sequencing on mice. These pathobionts worsen cancer-like features, affect T cells, and alter Myc-signaling and fatty acid metabolism. However, the study's focus on these two in the ApcMin/+ model limits its generalizability. Lack of long-term analysis, omission of both sexes, short progression time, and limited stats raise questions. If the authors can't address all queries, justification is required.

Reviewers' comments:

Reviewer's Responses to Questions

**Comments to the Author**

1. Is the manuscript technically sound, and do the data support the conclusions?

Reviewer #1: Yes

Reviewer #2: Yes

2. Has the statistical analysis been performed appropriately and rigorously? 

Reviewer #1: Yes

Reviewer #2: Yes

3. Have the authors made all data underlying the findings in their manuscript fully available?

Reviewer #1: Yes

Reviewer #2: Yes

4. Is the manuscript presented in an intelligible fashion and written in standard English?

Reviewer #1: Yes

Reviewer #2: Yes

5. Review Comments to the Author

Reviewer #1: The manuscript by Josh Jones et al., titled " Keystone pathobionts associated with colorectal cancer promote oncogenic reprograming " has been well written as well as studied manuscript which addresses the function of two pathobionts in colorectal cancer (CRC), Fusobacterium nucleatum (Fn) and enterotoxigenic Bacteroides fragilis (ETBF). To study the specific effects of these pathobionts on the gut epithelium and local immune system, the researchers performed single-cell RNA sequencing (scRNA-seq) on both wildtype mice and a mouse model of CRC. In the CRC animal model, Fn and ETBF were observed to worsen cancer-like transcriptional features in transit-amplifying and mature enterocytes. Furthermore, mice exposed to pathobionts had an increase in T cells, but these alterations weren't observed in the CRC mouse model. The researchers observed pathobiont-specific changes on Myc-signaling and fatty acid metabolism, indicating that Fn and ETBF contribute to carcinogenesis by promoting a cancer stem cell-like transit-amplifying and enterocyte population while impairing CTL cytotoxicity.

Limitation of the study: This study has several limitations that should be taken into consideration when interpreting the results. First, the scope of the experiments is limited to the effects of two specific pathobionts, Fn and ETBF, on intestinal cell composition and transcriptional profiles in ApcMin/+ mice. While this provides valuable insights into the impact of these pathobionts in a specific context, it does not encompass the full complexity of the microbiome or consider other potential factors contributing to colorectal cancer (CRC) development. Therefore, the findings may not be generalizable to all cases of CRC. Another important factor is lack of long-term analysis.

Minor Comment;

1) Why did the author choose to use ApcMin/+ mutant mice as the model for studying colorectal cancer, instead of other mouse models commonly used in CRC research?

2) Explain why does the limited focus on only two specific pathobionts, Fn and ETBF, in this study or investigating the effects on intestinal cell composition and transcriptional profiles in ApcMin/+ mice restrict the understanding of the full complexity of the microbiome and overlook other potential factors contributing to colorectal cancer development?

3) Explain to what extent does the use of ApcMin/+ mice, carrying a specific mutation associated with intestinal tumors, limit the ability to fully capture the complexity of human colorectal cancer? How do the inherent differences in physiology and tumor development between animal models and human disease impact the extrapolation of findings to human biology?

4) Discuses why author did not include both male and female mice in the study? How may the inclusion of sexes contribute to a more thorough knowledge of the effects of pathobiont exposure on intestinal cell composition and transcriptional patterns, considering the potential role of sex in the development of colorectal cancer?

5) Clarify why the author chose such a small-time span for the onset and progression of colorectal cancer. Why did the authors not focus on the substantially longer time point in their pathobiont exposure investigation?

6) The author mentions sample sizes (n ≥ 6 ApcMin/+ mice) but does not provide further details on statistical analyses. The significance of the observed differences and the generalizability of the findings would benefit from a comprehensive statistical analysis, explain?

7) Why did the author not include information about alterations in fatty acid metabolism in this study, despite it being mentioned in the study? Add heat map depicting the disruption of lipid metabolism gene based on the analysis conducted in this study?

Reviewer #2: Comments to the authors:

The manuscript entitled “Keystone pathobionts associated with colorectal cancer promote oncogenic reprograming” explores the role of 2 pathobionts in oncogenic reprograming in CRC employing scRNA-seq. The study is well conceptualized and executed with ample experiments and analyses. The manuscript is well written with appropriate and sufficient citations and the results are clearly explained. The findings are quite interesting and backed up by a reasonable amount of data. Although, these findings show a role of Fn and ETBF in potentiating tumorigenesis by influencing cancer associated signaling cascades, CRC initiation, and progression in the ApcMin/+ model, studies on other CRC related pathobionts are needed for diagnosis and therapeutics.

I recommend acceptance of the manuscript in its current form.

6. PLOS authors have the option to publish the peer review history of their article (what does this mean?). If published, this will include your full peer review and any attached files.

Reviewer #1: **Yes: **ZEESHAN AHMAD

Reviewer #2: No

---

## [Author Response · Author response to Decision Letter 0]

13 Nov 2023

Response to Reviewers

Reviewer #1: The manuscript by Josh Jones et al., titled " Keystone pathobionts associated with colorectal cancer promote oncogenic reprograming " has been well written as well as studied manuscript which addresses the function of two pathobionts in colorectal cancer (CRC), Fusobacterium nucleatum (Fn) and enterotoxigenic Bacteroides fragilis (ETBF). To study the specific effects of these pathobionts on the gut epithelium and local immune system, the researchers performed single-cell RNA sequencing (scRNA-seq) on both wildtype mice and a mouse model of CRC. In the CRC animal model, Fn and ETBF were observed to worsen cancer-like transcriptional features in transit-amplifying and mature enterocytes. Furthermore, mice exposed to pathobionts had an increase in T cells, but these alterations weren't observed in the CRC mouse model. The researchers observed pathobiont-specific changes on Myc-signaling and fatty acid metabolism, indicating that Fn and ETBF contribute to carcinogenesis by promoting a cancer stem cell-like transit-amplifying and enterocyte population while impairing CTL cytotoxicity.

Limitation of the study: This study has several limitations that should be taken into consideration when interpreting the results. First, the scope of the experiments is limited to the effects of two specific pathobionts, Fn and ETBF, on intestinal cell composition and transcriptional profiles in ApcMin/+ mice. While this provides valuable insights into the impact of these pathobionts in a specific context, it does not encompass the full complexity of the microbiome or consider other potential factors contributing to colorectal cancer (CRC) development. Therefore, the findings may not be generalizable to all cases of CRC. Another important factor is lack of long-term analysis.

We thank the reviewer for this thoughtful outline of our work and its limitations. We agree that our results provide a preliminary glimpse of how these organisms, often associated with CRC, may affect hosts with mutations in Apc, but have to be interpreted within the context of this mouse model. 

Minor Comment;

1) Why did the author choose to use ApcMin/+ mutant mice as the model for studying colorectal cancer, instead of other mouse models commonly used in CRC research?

The purpose of study was to determine the scope and magnitude of host change elicited perturbation with CRC pathobionts. To those ends, the ApcMin/+ model was chosen because the mice do not carry, outside of a mutant copy of Apc (the earliest and most common sporadic CRC mutation), any other driver oncogene mutations. Most importantly, we chose this model because it is extremely well-characterized and other pathobionts had been previously tested for their effects on pathogenesis in this model (described in lines 46-49 and 67-68), providing a basis for comparison. In this model, tumors develop spontaneously and the rate of tumor growth is not rapid. Therefore, transcriptomic changes and increased tumorigenesis captured are strongly associated with the pathobiont perturbation rather than oncogene induction. We do recognize that there are limitations in choosing this model, as it does not fully recapitulate the phenotypes observed in human CRC. However, other models, e.g. ApcMin mutants, Kras, p53 drivers, have similar shortcomings, either resulting in rapid tumorigenesis or being driven by oncogenes that can have a myriad of host responses downstream, making it hard to disentangle microbe-specific versus oncogene induced changes. 

2) Explain why does the limited focus on only two specific pathobionts, Fn and ETBF, in this study or investigating the effects on intestinal cell composition and transcriptional profiles in ApcMin/+ mice restrict the understanding of the full complexity of the microbiome and overlook other potential factors contributing to colorectal cancer development?

We focused our study on Fn and ETBF, given that they were among the most extensively studied CRC pathobionts at the time. These two organisms are consistently more prevalent in stool samples from CRC patients compared to controls, supporting their importance in disease initiation and/or progression. The pathobiology of Fn and ETBF is somewhat understood, with known molecular mechanisms impacting disease (outlined in lines 39-49, 54-59,and 342-348) enabling us to investigate the downstream effects. Furthermore, while we agree that it would be interesting to examine a greater number of factors, including microbiome composition, the number of mice to be able to examine effects of additional species within the microbiome using transcriptional read-outs is unfeasible.

3) Explain to what extent does the use of ApcMin/+ mice, carrying a specific mutation associated with intestinal tumors, limit the ability to fully capture the complexity of human colorectal cancer? How do the inherent differences in physiology and tumor development between animal models and human disease impact the extrapolation of findings to human biology?

The loss of heterozygosity at the Apc locus is considered an initiating or early event in human sporadic CRC, and the same holds true for ApcMin/+ mice. However, differences in tumor sites (the murine small intestine versus the human colon) constrain the direct applicability of these findings to human biology (discussed in lines 306-313). Nonetheless, our results bolster the idea that CRC-associated pathobionts possess tumor-promoting tendencies. These tendencies merit further investigation, especially given their increased prevalence in patients and their correlation with more severe disease outcomes.

4) Discuses why author did not include both male and female mice in the study? How may the inclusion of sexes contribute to a more thorough knowledge of the effects of pathobiont exposure on intestinal cell composition and transcriptional patterns, considering the potential role of sex in the development of colorectal cancer?

We did in fact use mice of both sexes. Differences in CRC based on sex are believed to be primarily associated with the sidedness of the tumor burden in humans. In ApcMin/+ mice, tumorigenesis doesn't reflect the same sidedness patterns observed in humans; murine tumors are largely confined to the small intestine, while human disease predominantly affects the distal and proximal colon. Other research groups have observed sex-specific differences in ApcMin/+ mice, although their data suggest that these effects are subtle and region-specific. By increasing our sample size and including both male and female subjects, we would be able to provide more clarity on this. Additionally, we randomized the selection of ApcMin/+ and WT mice in this study. https://doi.org/10.1073/pnas.1323064111. 

5) Clarify why the author chose such a small-time span for the onset and progression of colorectal cancer. Why did the authors not focus on the substantially longer time point in their pathobiont exposure investigation?

We selected a shorter time course because our primary interest lay in the direct role of these two pathobionts on the initial stages of pathobiont-associated tumorigenesis. The transcriptional signal of later time points may reflect cancer progression rather than initiation. We aimed to identify cell populations that initiate tumor formation early on. However, we chose a longer time point for tumor enumeration to understand the rate of gross tumorigenesis at a more advanced stage of the disease in the mice.

6) The author mentions sample sizes (n ≥ 6 ApcMin/+ mice) but does not provide further details on statistical analyses. The significance of the observed differences and the generalizability of the findings would benefit from a comprehensive statistical analysis, explain?

We initially contemplated conducting power analyses to determine the necessary sample size for our tumor burden experiments. However, estimating the rate of attrition and effect size in our experimental setup was challenging, making any calculations difficult. To address this, we conducted a literature search to determine the sample size typically used for similar tumor burden experiments. (https://doi.org/10.1186/s13046-020-01677-w and https://doi.org/10.1016/j.chom.2018.01.007). We started with 10 mice per group. However, attrition due to anemia or meeting ethical end points led to fewer mice per group. Note that the number of mice lost due to attrition was roughly equal across groups. 

7) Why did the author not include information about alterations in fatty acid metabolism in this study, despite it being mentioned in the study? Add heat map depicting the disruption of lipid metabolism gene based on the analysis conducted in this study?

Supplemental Figure 1showing the top differentially enriched pathways in the transcriptomes of CSC-like TA cells from the Fn-exposed ApcMin/+ mouse compared with the PBS-treated ApcMin/+ mouse shows fatty acid metabolism as the top hit. We propose that fatty acid metabolism is upregulated in these cancer-stem cell-like cells to meet the elevated metabolic demands of rapidly dividing cancer cells, a finding other groups have noted.

Reviewer #2: Comments to the authors:

The manuscript entitled “Keystone pathobionts associated with colorectal cancer promote oncogenic reprograming” explores the role of 2 pathobionts in oncogenic reprograming in CRC employing scRNA-seq. The study is well conceptualized and executed with ample experiments and analyses. The manuscript is well written with appropriate and sufficient citations and the results are clearly explained. The findings are quite interesting and backed up by a reasonable amount of data. Although, these findings show a role of Fn and ETBF in potentiating tumorigenesis by influencing cancer associated signaling cascades, CRC initiation, and progression in the ApcMin/+ model, studies on other CRC related pathobionts are needed for diagnosis and therapeutics.

I recommend acceptance of the manuscript in its current form.

Thank you for your interest and support of our manuscript!

---

## [Editor Report · Decision Letter 1]

14 Dec 2023

PONE-D-23-17692R1Keystone pathobionts associated with colorectal cancer promote oncogenic reprogramingPLOS ONE

Dear Dr. Brito,

Thank you for submitting your manuscript to PLOS ONE. After careful consideration, we feel that it has merit but does not fully meet PLOS ONE’s publication criteria as it currently stands. Therefore, we invite you to submit a revised version of the manuscript that addresses the points raised during the review process.

To further enhance the transparency and comprehensiveness of your manuscript, I have a few suggestions for additional details in the Ethical Considerations section:

Your initial statement about veterinary care is appropriate. However, it would be beneficial to expand on this by specifying the criteria and procedures used for daily monitoring of the mice. Additionally, please describe how any signs of distress or ill health were managed and addressed.

Given the focus of your study, a brief explanation regarding the decision not to quantify total tumor volumes is necessary. Elaborating on how tumor growth was monitored and managed within the ethical guidelines will significantly strengthen this section of your manuscript.

These additions will greatly enhance the ethical transparency and integrity of your manuscript.

We look forward to receiving your revised manuscript.

Kind regards,

Syed M. Faisal, Ph.D.

Academic Editor

PLOS ONE
---

## [Author Response · Author response to Decision Letter 1]

8 Jan 2024

We already submitted these in our previous review: 

Response to Reviewers

Reviewer #1: The manuscript by Josh Jones et al., titled " Keystone pathobionts associated with colorectal cancer promote oncogenic reprograming " has been well written as well as studied manuscript which addresses the function of two pathobionts in colorectal cancer (CRC), Fusobacterium nucleatum (Fn) and enterotoxigenic Bacteroides fragilis (ETBF). To study the specific effects of these pathobionts on the gut epithelium and local immune system, the researchers performed single-cell RNA sequencing (scRNA-seq) on both wildtype mice and a mouse model of CRC. In the CRC animal model, Fn and ETBF were observed to worsen cancer-like transcriptional features in transit-amplifying and mature enterocytes. Furthermore, mice exposed to pathobionts had an increase in T cells, but these alterations weren't observed in the CRC mouse model. The researchers observed pathobiont-specific changes on Myc-signaling and fatty acid metabolism, indicating that Fn and ETBF contribute to carcinogenesis by promoting a cancer stem cell-like transit-amplifying and enterocyte population while impairing CTL cytotoxicity.

Limitation of the study: This study has several limitations that should be taken into consideration when interpreting the results. First, the scope of the experiments is limited to the effects of two specific pathobionts, Fn and ETBF, on intestinal cell composition and transcriptional profiles in ApcMin/+ mice. While this provides valuable insights into the impact of these pathobionts in a specific context, it does not encompass the full complexity of the microbiome or consider other potential factors contributing to colorectal cancer (CRC) development. Therefore, the findings may not be generalizable to all cases of CRC. Another important factor is lack of long-term analysis.

We thank the reviewer for this thoughtful outline of our work and its limitations. We agree that our results provide a preliminary glimpse of how these organisms, often associated with CRC, may affect hosts with mutations in Apc, but have to be interpreted within the context of this mouse model. 

Minor Comment;

1) Why did the author choose to use ApcMin/+ mutant mice as the model for studying colorectal cancer, instead of other mouse models commonly used in CRC research?

The purpose of study was to determine the scope and magnitude of host change elicited perturbation with CRC pathobionts. To those ends, the ApcMin/+ model was chosen because the mice do not carry, outside of a mutant copy of Apc (the earliest and most common sporadic CRC mutation), any other driver oncogene mutations. Most importantly, we chose this model because it is extremely well-characterized and other pathobionts had been previously tested for their effects on pathogenesis in this model (described in lines 46-49 and 67-68), providing a basis for comparison. In this model, tumors develop spontaneously and the rate of tumor growth is not rapid. Therefore, transcriptomic changes and increased tumorigenesis captured are strongly associated with the pathobiont perturbation rather than oncogene induction. We do recognize that there are limitations in choosing this model, as it does not fully recapitulate the phenotypes observed in human CRC. However, other models, e.g. ApcMin mutants, Kras, p53 drivers, have similar shortcomings, either resulting in rapid tumorigenesis or being driven by oncogenes that can have a myriad of host responses downstream, making it hard to disentangle microbe-specific versus oncogene induced changes. 

2) Explain why does the limited focus on only two specific pathobionts, Fn and ETBF, in this study or investigating the effects on intestinal cell composition and transcriptional profiles in ApcMin/+ mice restrict the understanding of the full complexity of the microbiome and overlook other potential factors contributing to colorectal cancer development?

We focused our study on Fn and ETBF, given that they were among the most extensively studied CRC pathobionts at the time. These two organisms are consistently more prevalent in stool samples from CRC patients compared to controls, supporting their importance in disease initiation and/or progression. The pathobiology of Fn and ETBF is somewhat understood, with known molecular mechanisms impacting disease (outlined in lines 39-49, 54-59,and 342-348) enabling us to investigate the downstream effects. Furthermore, while we agree that it would be interesting to examine a greater number of factors, including microbiome composition, the number of mice to be able to examine effects of additional species within the microbiome using transcriptional read-outs is unfeasible.

3) Explain to what extent does the use of ApcMin/+ mice, carrying a specific mutation associated with intestinal tumors, limit the ability to fully capture the complexity of human colorectal cancer? How do the inherent differences in physiology and tumor development between animal models and human disease impact the extrapolation of findings to human biology?

The loss of heterozygosity at the Apc locus is considered an initiating or early event in human sporadic CRC, and the same holds true for ApcMin/+ mice. However, differences in tumor sites (the murine small intestine versus the human colon) constrain the direct applicability of these findings to human biology (discussed in lines 306-313). Nonetheless, our results bolster the idea that CRC-associated pathobionts possess tumor-promoting tendencies. These tendencies merit further investigation, especially given their increased prevalence in patients and their correlation with more severe disease outcomes.

4) Discuses why author did not include both male and female mice in the study? How may the inclusion of sexes contribute to a more thorough knowledge of the effects of pathobiont exposure on intestinal cell composition and transcriptional patterns, considering the potential role of sex in the development of colorectal cancer?

We did in fact use mice of both sexes. Differences in CRC based on sex are believed to be primarily associated with the sidedness of the tumor burden in humans. In ApcMin/+ mice, tumorigenesis doesn't reflect the same sidedness patterns observed in humans; murine tumors are largely confined to the small intestine, while human disease predominantly affects the distal and proximal colon. Other research groups have observed sex-specific differences in ApcMin/+ mice, although their data suggest that these effects are subtle and region-specific. By increasing our sample size and including both male and female subjects, we would be able to provide more clarity on this. Additionally, we randomized the selection of ApcMin/+ and WT mice in this study. https://doi.org/10.1073/pnas.1323064111. 

5) Clarify why the author chose such a small-time span for the onset and progression of colorectal cancer. Why did the authors not focus on the substantially longer time point in their pathobiont exposure investigation?

We selected a shorter time course because our primary interest lay in the direct role of these two pathobionts on the initial stages of pathobiont-associated tumorigenesis. The transcriptional signal of later time points may reflect cancer progression rather than initiation. We aimed to identify cell populations that initiate tumor formation early on. However, we chose a longer time point for tumor enumeration to understand the rate of gross tumorigenesis at a more advanced stage of the disease in the mice.

6) The author mentions sample sizes (n ≥ 6 ApcMin/+ mice) but does not provide further details on statistical analyses. The significance of the observed differences and the generalizability of the findings would benefit from a comprehensive statistical analysis, explain?

We initially contemplated conducting power analyses to determine the necessary sample size for our tumor burden experiments. However, estimating the rate of attrition and effect size in our experimental setup was challenging, making any calculations difficult. To address this, we conducted a literature search to determine the sample size typically used for similar tumor burden experiments. (https://doi.org/10.1186/s13046-020-01677-w and https://doi.org/10.1016/j.chom.2018.01.007). We started with 10 mice per group. However, attrition due to anemia or meeting ethical end points led to fewer mice per group. Note that the number of mice lost due to attrition was roughly equal across groups. 

7) Why did the author not include information about alterations in fatty acid metabolism in this study, despite it being mentioned in the study? Add heat map depicting the disruption of lipid metabolism gene based on the analysis conducted in this study?

Supplemental Figure 1showing the top differentially enriched pathways in the transcriptomes of CSC-like TA cells from the Fn-exposed ApcMin/+ mouse compared with the PBS-treated ApcMin/+ mouse shows fatty acid metabolism as the top hit. We propose that fatty acid metabolism is upregulated in these cancer-stem cell-like cells to meet the elevated metabolic demands of rapidly dividing cancer cells, a finding other groups have noted.

Reviewer #2: Comments to the authors:

The manuscript entitled “Keystone pathobionts associated with colorectal cancer promote oncogenic reprograming” explores the role of 2 pathobionts in oncogenic reprograming in CRC employing scRNA-seq. The study is well conceptualized and executed with ample experiments and analyses. The manuscript is well written with appropriate and sufficient citations and the results are clearly explained. The findings are quite interesting and backed up by a reasonable amount of data. Although, these findings show a role of Fn and ETBF in potentiating tumorigenesis by influencing cancer associated signaling cascades, CRC initiation, and progression in the ApcMin/+ model, studies on other CRC related pathobionts are needed for diagnosis and therapeutics.

I recommend acceptance of the manuscript in its current form.

Thank you for your interest and support of our manuscript!

---

## [Editor Report · Decision Letter 2]

16 Jan 2024

Keystone pathobionts associated with colorectal cancer promote oncogenic reprograming

PONE-D-23-17692R2

Dear Dr. Brito,

We’re pleased to inform you that your manuscript has been judged scientifically suitable for publication and will be formally accepted for publication once it meets all outstanding technical requirements.

Kind regards,

Syed M. Faisal, Ph.D.

Academic Editor

PLOS ONE
---

## [Editor Report · Acceptance letter]

9 Feb 2024

PONE-D-23-17692R2 

PLOS ONE

Dear Dr. Brito, 

I'm pleased to inform you that your manuscript has been deemed suitable for publication in PLOS ONE. Congratulations! Your manuscript is now being handed over to our production team.

Kind regards, 

on behalf of

Dr. Syed M. Faisal 

Academic Editor

PLOS ONE